# Indigenous Yeast, Lactic Acid Bacteria, and Acetic Acid Bacteria from Cocoa Bean Fermentation in Indonesia Can Inhibit Fungal-Growth-Producing Mycotoxins

**Endang Sutriswati Rahayu [1,2,\*], Rokhmat Triyadi [3], Rosyida N. B. Khusna [1], Titiek Farianti Djaafar [4,\*], Tyas Utami [1], Tri Marwati [4] and Retno Utami Hatmi [4]**

[1] Department of Food and Agricultural Product Technology, Universitas Gadjah Mada, Yogyakarta 55281, Indonesia; rosyidanbk@mail.ugm.ac.id (R.N.B.K.); tyas_utami@ugm.ac.id (T.U.)
[2] Centre for Food and Nutrition Studies, Universitas Gadjah Mada, Yogyakarta 55281, Indonesia
[3] Provincial Office of Drug and Food Control, Bali 80234, Indonesia; pomdenpasar@yahoo.co.id
[4] Department of Postharvest, Assessment Institute for Agricultural Technology, Yogyakarta 55584, Indonesia; watipasca2@gmail.com (T.M.); bptpyogya@yahoo.com (R.U.H.)
\* Correspondence: endangsrahayu@ugm.ac.id (E.S.R.); titiekfd1212@gmail.com (T.F.D.); Tel.: +62-812-2792-5997 (T.F.D.)

**Abstract:** Cocoa bean fermentation is an important process in the manufacturing of cocoa products. It involves microbes, such as lactic acid bacteria, yeast, and acetic acid bacteria. The presence of mold in cocoa bean fermentation is undesired, as it reduces the quality and may produce mycotoxins, which can cause poisoning and death. *Aspergillus niger* is a fungus that produces ochratoxin A, which is often found in dried agricultural products such as seeds and cereals. In this study, we applied indigenous *Candida famata* HY-37, *Lactobacillus plantarum* HL-15, and *Acetobacter* spp. HA-37 as starter cultures for cocoa bean fermentation. We found that the use of *L. plantarum* HL-15 individually or in combination *Candida famata* HY-37, *Lactobacillus plantarum* HL-15, and *Acetobacter* spp. HA-37 as a starter for cocoa bean fermentation can inhibit the growth of *A. niger* YAC-9 and the synthesis of ochratoxin A during fermentation and drying. With biological methods that use indigenous *Lactobacillus plantarum* HL-15 individually or in combination with *Candida famata* HY-37 and *Acetobacter* spp. HA-37, we successfully inhibited contamination by ochratoxin-A-producing fungi. Thus, the three indigenous microbes should be used in cocoa bean fermentation to inhibit the growth of fungi that produce mycotoxins and thus improve the quality.

**Keywords:** *Candida famata* HY-37; *Lactobacillus plantarum* HL-15; *Acetobacter* spp. HA-37; cocoa bean fermentation; anti-fungal growth; ochratoxin A

## 1. Introduction

Quality fermented cocoa (*Theobroma cacao* Linn.) beans are the main raw materials in the processing of chocolate. The cocoa bean fermentation process plays an important role in the production of quality cocoa beans for various chocolate products and other products made from cocoa beans. During cocoa bean fermentation, the mucilaginous pulp is removed, thus preventing germination and initiating the development of the aroma, flavor, and color [1–4].

Successful fermentation of cocoa beans is determined by three microbes, namely yeast, lactic acid bacteria (LAB), and acetic acid bacteria (AAB), which grow naturally. Under the anaerobic conditions of the first 24 h of fermentation, yeast is dominant. Then, the fermentation continues, and the pulp is drained such that air can enter the mass of the cocoa beans. In these conditions, LAB and AAB grow well after about 24–72 h of fermentation [2,5,6].

In unfavorable fermentation conditions, fungi are found, causing negative effects on the quality and safety cocoa beans [7–9]. Certain fungal species can produce mycotoxins, such as ochratoxin A [10–12]. Several studies have been conducted in an attempt to reduce

contamination by mycotoxin-producing fungi. The detoxification methods conducted are chemical, physical [13–15], and biological methods [16–18]. The use of biological methods with antagonistic microbes is considered safer than the use of chemical and physical methods. Lactic acid bacteria are a type of bacteria that can be used in biological methods to prevent contamination with ochratoxin A by inhibiting fungal growth and ochratoxin A synthesis [8].

The presence of bacteria that produce lactic acid (LAB) in the fermentation of cocoa beans has potential inhibitory properties against mycotoxin-producing fungi [8,19,20]. Yeast also has potential as an antifungal agent because it produces proteinaceous metabolites [8]. Therefore, it is necessary to conduct research that aims to find isolates of the indigenous LAB of fermented cocoa beans that have potential as antifungal agents against fungi that produce mycotoxins. Indigenous lactic acid bacteria were isolated from the fermentation of cocoa beans in the Gunungkidul district, Yogyakarta [21]. The LAB identified as *Lactobacillus plantarum* HL-15 has antifungal activity against *A. niger* YAC-9. At the same time, the authors of [22] also isolated yeast and acetic acid bacteria from fermented cocoa beans in the Gunungkidul Regency, Yogyakarta. Thus, in this study, we report that the abilities of indigenous *Candida famata* HY-37, *L. plantarum* HL-15, and *Acetobacter* spp. HA-37 to inhibit fungal growth that results in the production of ochratoxin A. With biological methods that use the indigenous LAB individually or in combination with yeast and AAB, we have successfully inhibited contamination with ochratoxin-A-producing fungi.

## 2. Materials and Methods

### 2.1. Media and Equipment

The media used were Standard Plate Count Agar (APHA) (OXOID CM0463 Basingstoke, England), Dichloran Rose Bengal Chloramphenicol Agar (DRBC) (MERCK Darmstadt, Germany), de Man Ragosa Sharpe broth (MRSB) (MERCK Darmstadt, Germany), Bacteriological Peptone (OXOID LP0037 Basingstoke, England), D (+)-glucose (MERCK Darmstadt, Germany), yeast extract (OXOID LP0021 Basingstoke, England) and malt extract broth (MEB) (MERCK Darmstadt, Germany), bacteriological Agar (OXOID LP0011 Basingstoke, England), calcium carbonate (CaCO$_3$) (MERCK Darmstadt, Germany), 0.85% NaCl solution, 3% H$_2$O$_2$ solution, 0.32% CaCO$_3$ solution, 20% glycerol solution, and skim milk. The equipment used included luminaire flow, petri dishes, 0.1 and 1 mL micropipettes, conical tube, Erlenmeyer flask, Quebec colony counter, stomacher, vortex, shaker, incubator, microscope, ELISA test kit (AgraQuant®ELISA Ochratoxin Assay 2/40) (ROMER Labs, Singapore, Singapore), and spectrophotometer.

### 2.2. Sample and Microorganisms

Cocoa fruits of the Lindak variety were obtained from the Sido Dadi farmer group (Gunungkidul Regency, Yogyakarta) for the fermentation assessment. The indigenous *L. plantarum* HL-15 LAB, *Candida famata* HY-37 yeast, *Acetobacter* spp. HA-37 AAB are microbes that were isolated from cocoa bean fermentation in previous research [21,22] and were maintained in the Center for Food and Nutrition Studies (FNCC), Universitas Gadjah Mada, Yogyakarta. The *A. niger* YAC-9 fungus was used as an ochratoxin A producer [23] and was obtained from the FNCC.

### 2.3. Preparation of Microorganisms for Cocoa Bean Fermentation

The media for *C. famata* HY-37 were malt extract broth (20 g/L); MRS broth (52 g/L) was used for *L. plantarum* HL-15; peptone (4,5 g/L), glucose (20 g/L), and yeast extract (4,5 g/L) were used for *Acetobacter* spp. HA-37. Culture rejuvenation was performed by taking 1–2 of the cultures, placing them into the growing media, and then incubating at 37 °C for 24 h. Furthermore, 5 mL of the cultures were transferred to an Erlenmeyer flask containing 50 mL of each growing medium and incubated at 37 °C for 24 h. The culture

starter was stored in a cool room until it was used. The concentration of the starter cultures used for cocoa bean fermentation was $10^9$ CFU/mL.

The *A. niger* YAC-9 culture was grown in an MEA slant tube at 25 °C for 7 days. Spores were harvested using 0.1% Tween 80; thus, the final concentration was $10^9$ CFU/mL [9].

### 2.4. Fermentation of Cocoa Beans

Cocoa bean fermentation was performed with the following four treatments: (1) natural fermentation; (2) fermentation through the addition of *L. plantarum* HL-15 and *A. niger* YAC-9; (3) natural fermentation through the addition of *A. niger* YAC-9; (4) fermentation through the addition of mixed starter cultures of *L. plantarum* HL-15, *C. famata* HY-37, *Acetobacter* spp. HA-37, and *A. niger* YAC-9.

A total of 2.5 kg of cocoa beans were placed into a bamboo basket, and with the starter culture was added according to the treatment. As much as 25 mL of the starter culture was added, with a total culture of $10^9$ CFU/mL. The top of the bamboo basket was covered with banana leaves and burlap sacks. Then, the fermentation of cocoa beans was conducted at room temperature for 5 days and reserved for two days. After fermentation, the cocoa beans were immersed for 1 h and then washed. Furthermore, the fermented cocoa beans were sun dried in a greenhouse. The drying was conducted for 5 days until the water content of the cocoa beans reached 7–8%.

The analysis was performed every day during the fermentation and drying; it included an analysis of the pH, temperature, and total microbial count of the LAB, AAB, yeast, and *A. niger* YAC-9. Total microbial count analysis was conducted by using the pour-plate method [24]. The analysis of pH was conducted by using pH meter [6].

### 2.5. Extraction and Analysis of Ochratoxin A

The analysis of ochratoxin A was conducted based on the method from [12] with modifications during the fermentation and drying. The sampling for the ochratoxin A analysis was conducted during the fermentation and drying (days 0, 1, 5, and 10). Twenty-gram samples were blended, transferred into 250 mL Erlenmeyer flask, added with 100 mL methanol: demineralized water (7:3, *v/v*), and then shaken for 3 min. The solution was filtered using by Whatman filter paper number 1. A total of 2 mL of filtrate was taken and stored in an Eppendorf vial. The ochratoxin A produced by *A. niger* YAC-9 was measured with a solid-phase direct enzyme immunoassay using ELISA test kit, the AgraQuant® ELISA Ochratoxin Assay 2/40, which is designed for cocoa products (ROMER Labs, Singapore, Singapore).

### 2.6. Statistical Analysis

All experiments were conducted in triplicate, and the results are reported as mean $\pm$ standard deviation (SD, n = 3).

## 3. Results

### 3.1. Cocoa Bean Fermentation

The microbial growth patterns and changes in the pH and temperature of all treatments for cocoa bean fermentation can be seen in Figures 1–4. At the beginning of fermentation, the growth of the yeast increased, followed by the growth of lactic acid bacteria, and these began to decline on day 2 until the end of fermentation (day 5).

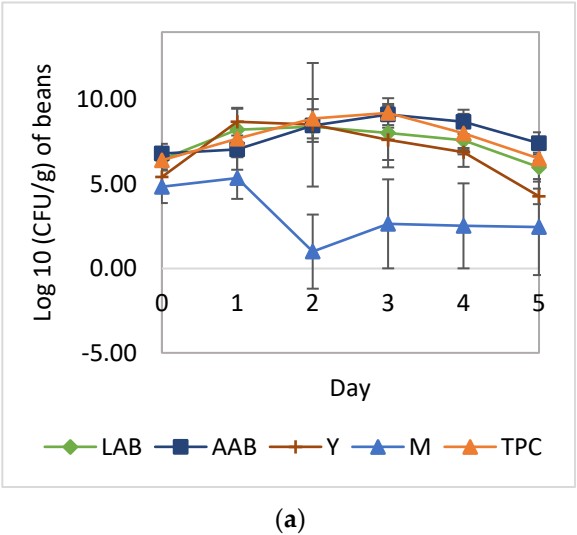
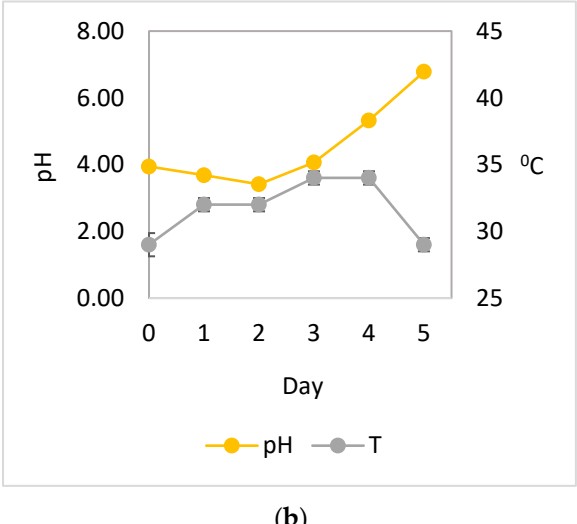

(**a**)  (**b**)

**Figure 1.** Microbial growth patterns and changes in pH and temperature during the natural fermentation of cocoa beans: (**a**) growth of yeast, LAB, AAB, and mold; (**b**) temperature and pH. LAB, total lactic acid bacteria; AAB, total acetic acid bacteria; Y, total yeast; M, total mold; TPC, total plate count; pH, pH pulp; T, temperature.

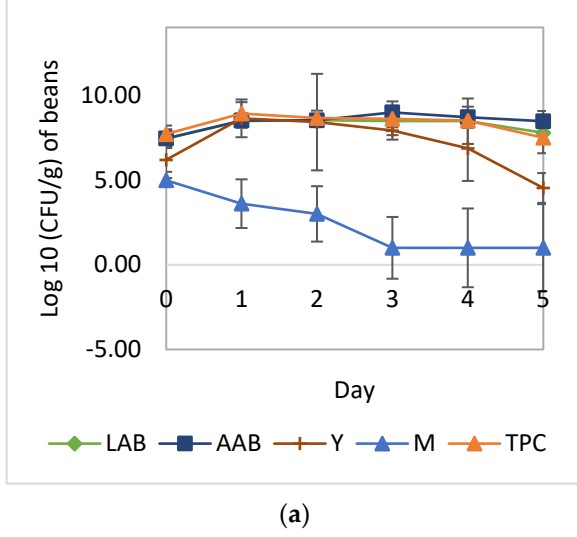
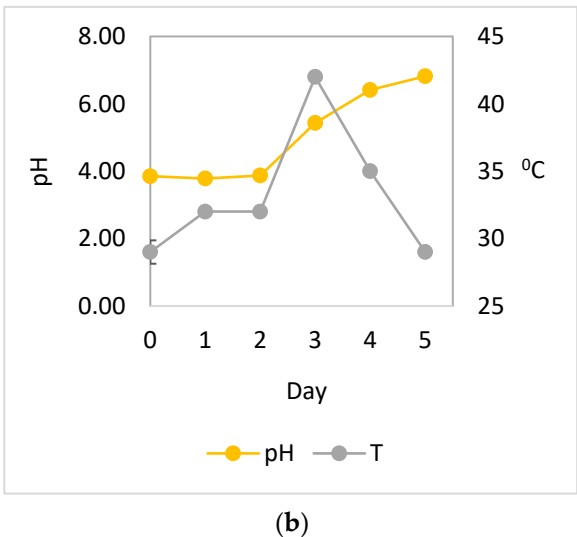

(**a**)  (**b**)

**Figure 2.** Microbial growth patterns and changes in pH and temperature during cocoa bean fermentation with the addition of *L. plantarum* HL-15 and *A. niger* YAC-9: (**a**) growth of yeast, LAB, AAB, and mold during fermentation; (**b**) temperature and pH. LAB, total lactic acid bacteria; AAB, total acetic acid bacteria; Y, total yeast; M, total mold; TPC, total plate count; pH, pH pulp; T, temperature.

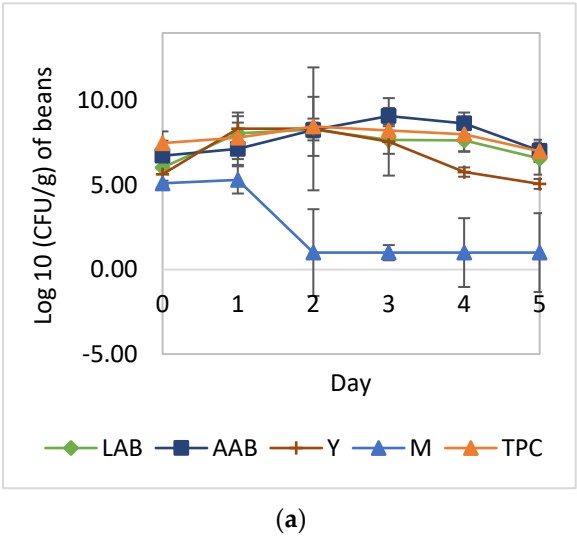 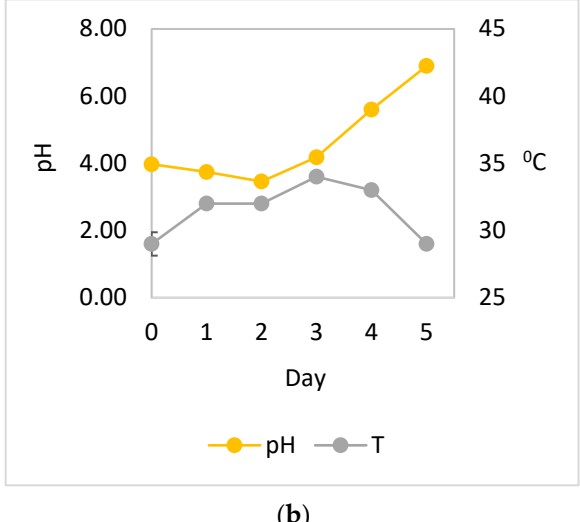

(**a**)  (**b**)

**Figure 3.** Microbial growth patterns and changes in pH and temperature during the natural fermentation of cocoa beans with the addition of *A. niger* YAC-9: (**a**) growth of yeast, LAB, AAB, and mold; (**b**) temperature and pH. LAB, total lactic acid bacteria; AAB, total acetic acid bacteria; Y, total yeast; M, total mold; TPC, total plate count; pH, pH pulp; T, temperature.

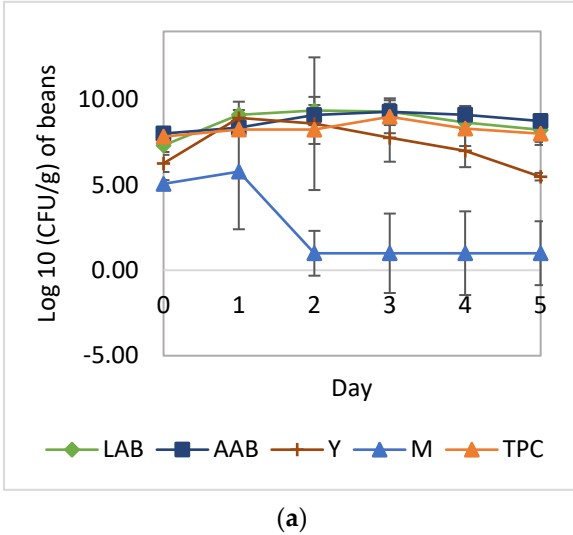 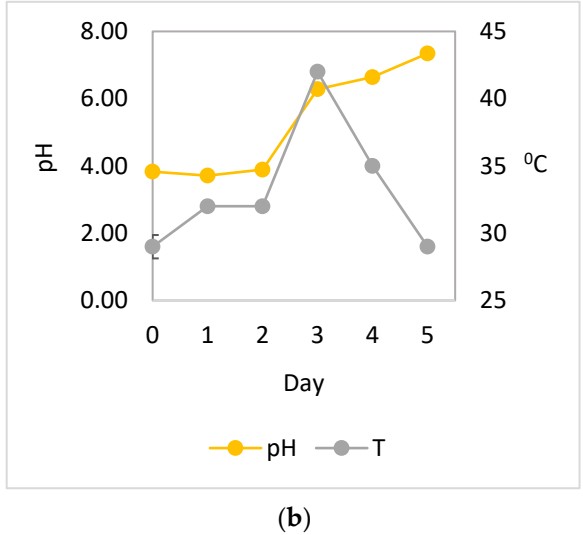

(**a**)  (**b**)

**Figure 4.** Microbial growth patterns and changes in pH and temperature during the fermentation of cocoa beans with the addition of *L. plantarum* HL-15, *C. famata* HY-37, *Acetobacter* spp. HA-37, and *A. niger* YAC-9: (**a**) growth of yeast, LAB, AAB, and mold; (**b**) temperature and pH. LAB, total lactic acid bacteria; AAB, total acetic acid bacteria; Y, total yeast; M, total mold; TPC, total plate count; pH, pH pulp; T, temperature.

The cocoa bean fermentation with the addition of *L. plantarum* HL-15 as a starter culture individually or in combination with *C. famata* HY-37 and *Acetobacter* spp. HA-37 was able to inhibit the growth of *A. niger* YAC-9 (Figures 2 and 4). *A. niger* YAC-9, which was added to the spontaneous cocoa bean fermentation process, was also able to inhibit growth (Figure 3). The growth of *A. niger* YAC-9 could be inhibited by as much as 1 log. This shows that in spontaneous fermentation, there is the growth of yeast, BAL, and AAB, which causes the temperature in the fermentation box to increase such that the fungus does not grow.

The addition of LAB individually or in a mixed culture with yeast and acetic acid bacteria (Figures 2 and 4) resulted in a greater increase in temperature compared to fermen-

tation without the addition of all three cultures—*L. plantarum* HL-15, *C. famata* HY-37, and *Acetobacter* spp. HA-37 (Figures 1 and 3).

### 3.2. Drying of Fermented Cocoa Beans

The growth patterns of *C. famata* HY-37, *L. plantarum* HL-15, *Acetobacter* spp. HA-37, and *A. niger* YAC-9 can be seen in Figure 5. The use of *L. plantarum* HL-15 as a starter culture was able to inhibit mold growth during fermentation and drying. The population of *A. niger* YAC-9 increased on drying days 8 and 9, but at the end of the drying, it decreased by about 4 logs for the cocoa beans fermentation with *L. plantarum* HL-15 or in combination with *C. famata* HY-37 and *Acetobacter* spp. HA-37 (Figure 5b,d). During the cocoa bean fermentation without the addition of *C. famata* HY-37, *L. plantarum* HL-15, and *Acetobacter* spp. HA-37 (Figure 5a,c), the *A. niger* YAC-9 population continued to increase until the end of drying (day 10).

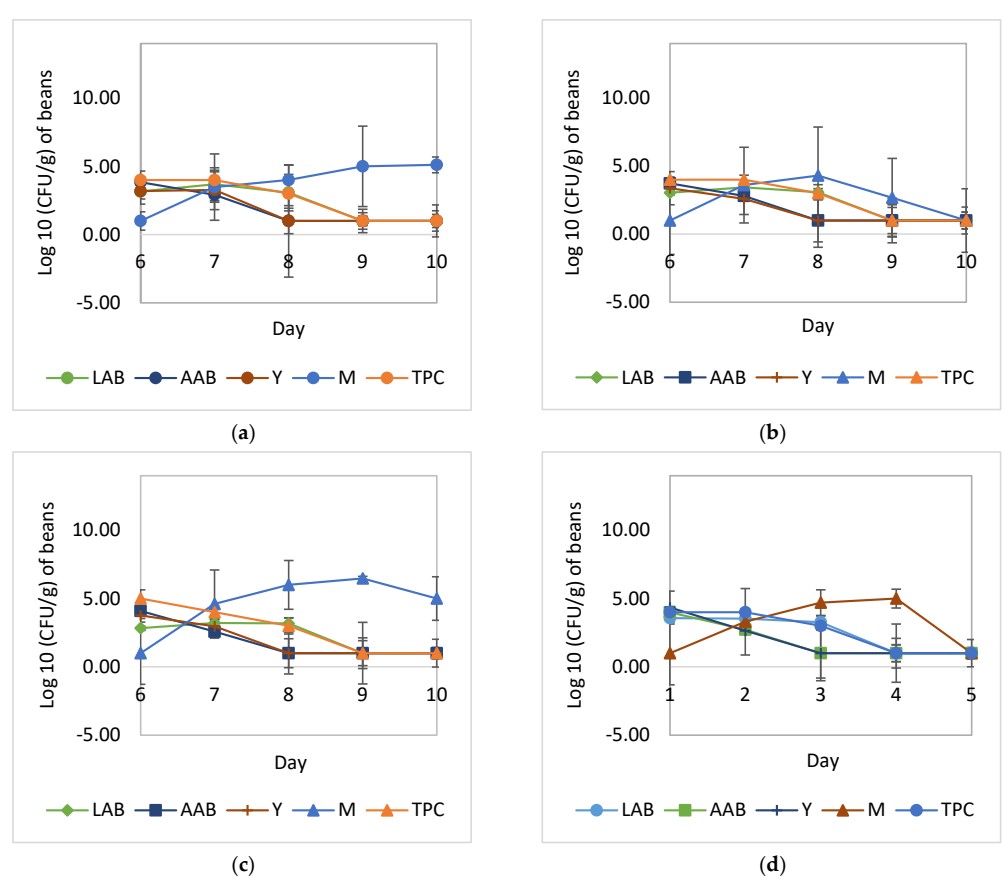

(a)

(b)

(c)

(d)

**Figure 5.** Growth of yeast, LAB, AAB, and mold during the course of drying: (**a**) natural fermentation; (**b**) fermentation with the addition of *L. plantarum* HL-15 and *A. niger* YAC-9; (**c**) natural fermentation with the addition of *A. niger* YAC-9; (**d**) fermentation with the addition of *L. plantarum* HL-15, *C. famata* HY-37, *Acetobacter* spp. HA-37, and *Aspergillus niger* YAC-9. LAB, total lactic acid bacteria; AAB, total acetic acid bacteria; Y, total yeast; M, total mold; TPC, total plate count.

### 3.3. Ochratoxin A Level in Fermented Cocoa Beans

The concentration of ochratoxin A (OTA) during fermentation and drying is presented in Figure 6. During fermentation, the synthesis of OTA occurred, but with the addition of a culture starter, the synthesis of OTA may be inhibited (Figure 6b,d). This is consistent with the inhibition of the growth of *A. niger* YAC-9 through the addition of *L. plantarum* HL-15 individually or in combination with *C. famata* HY-37 and *Acetobacter* spp. HA-37 (Figures 2a and 4a). Similarly, during the drying process, the OTA synthesis was inhibited by the presence of *L. plantarum* HL-15.

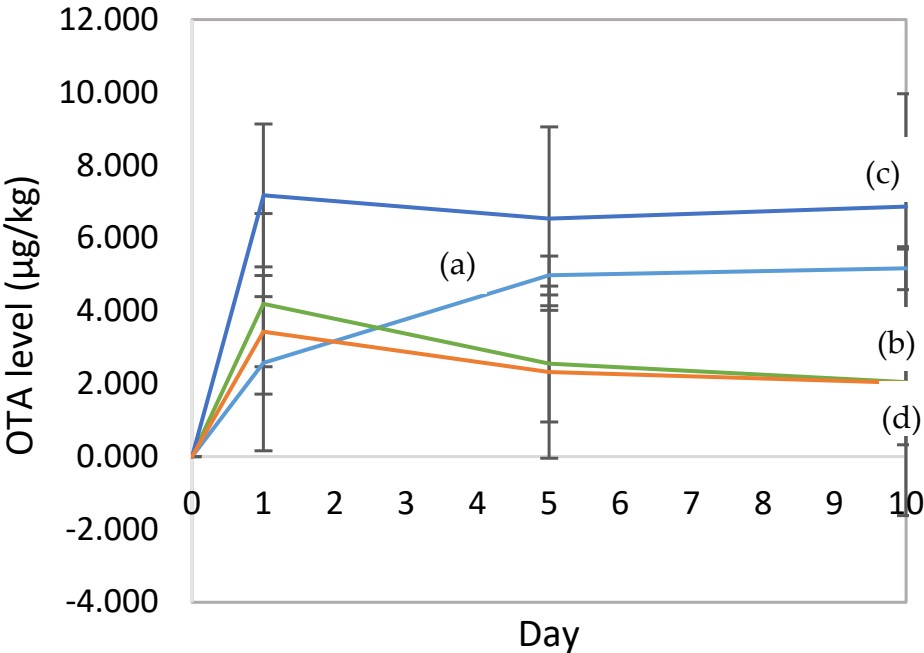

**Figure 6.** The OTA concentration during fermentation (days 0, 1, and 5) and drying (day 10):
(**a**) natural fermentation; (**b**) fermentation of cocoa beans with the addition of *L. plantarum* HL-15 and
*A. niger* YAC-9 as a contaminant; (**c**) fermentation of cocoa beans with the addition of *A. niger* YAC-9
as a contaminant; (**d**) cocoa bean fermentation with the addition of a mixed starter culture consisting
of *L. plantarum* HL-15, *C. famata* HY-37, *Acetobacter* spp. HA-37, and *A. niger* YAC-9 as a contaminant.
(**b**,**d**) show lower levels of ochratoxin contamination than those of (**a**,**d**). This shows that there was a
good effect from the addition of the LAB.

The OTA content during the fermentation with the addition of *A. niger* YAC-9 as
a contaminant had a different pattern (Figure 6c); the OTA content on the first day of
fermentation reached 7.17 µg/kg. This yield was considerably greater than that of the
cocoa beans in the other treatments.

In this study, the initial pH of the cocoa pulp was about 4.0. On day 2 of fermentation,
the pH decreased due to the formation of organic acids due to the microbial metabolism;
along with the decrease in pH, there was a decrease in the OTA concentration (Figure 6).

The external appearance of the dry cocoa beans in this treatment showed fungal
contamination due to the appearance of fungal mycelia adhering to the surface of the beans
and the discoloration of the beans (Figure 7c). No OTA was detected in the cocoa pulp
before fermentation. The ochratoxin A was formed consistent with the growth of *A. niger*
YAC-9, as seen in Figures 1–5.

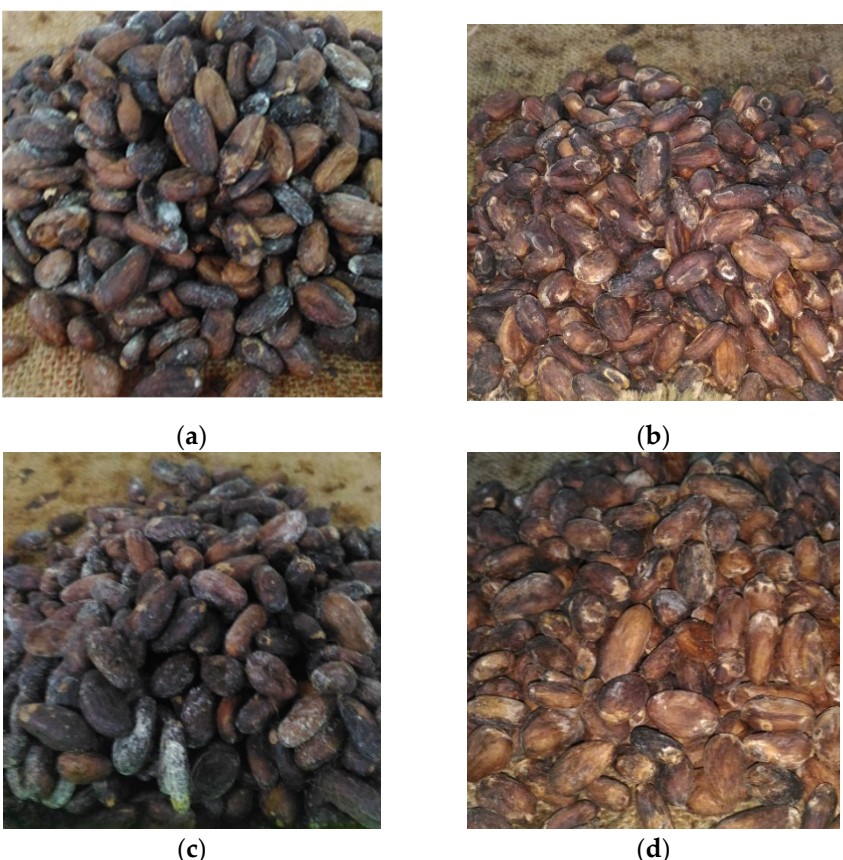

**Figure 7.** The appearance of fungal mycelia on the surfaces of the dry cocoa beans (days 10): (**a**) natural fermentation; (b) fermentation of cocoa beans with the addition of *L. plantarum* HL-15 and *A. niger* YAC-9; (**c**) cocoa bean fermentation with the addition of *A. niger* YAC-9; (d) cocoa bean fermentation with the addition of a mixed starter consisting of *L. plantarum* HL-15, *C. famata* HY-37, *Acetobacter* spp. HA-37, and *A. niger* YAC-9. There were no fungal mycelia in the treatments in (**b,d**). The color of the fermented cocoa beans that were contaminated by *A. niger* was blackish brown, as shown in (**a,c**).

## 4. Discussion

Cocoa bean fermentation plays an important role in the production of quality cocoa beans. The three main microbes involved in the fermentation of cocoa beans are yeast, LAB, and AAB [2,5,25–27]. Fungal contamination is the main problem in the fermentation of cocoa beans by cocoa farmers. Fungal contamination produces toxins that cause fermented cocoa beans to be unsafe when used as a raw material for the chocolate processing industry. In addition, it causes fermented cocoa beans to have a low quality, especially if there are fungal mycelia on the surfaces of the cocoa beans. Therefore, we conducted cocoa bean fermentation with the addition of yeast, LAB, and AAB, which are known to have antifungal properties. The LAB had the ability to inhibit the growth of fungi throughout the fermentation and drying of the cocoa beans [6,20,28].

The growth patterns of yeast, LAB, and AAB in this study are in accordance with what has been conveyed in previous studies. At the beginning of fermentation, yeast growth increased, followed by the growth of lactic acid bacteria; these began to decline from day 2 until the end of fermentation (day 5). However, the growth of AAB increased after two days of fermentation and decreased from the fourth to the fifth day. This is consistent with the results of research conducted by other authors [4,5]. At the beginning of fermentation, a depectinization reaction occurred due to the yeast; thus, the viscosity of the pulp decreased, allowing air to enter the mass of the cocoa beans [25]. Aeration encourages the growth of AAB and suppresses the growth of yeast [4]. Lactic acid bacteria metabolize

glucose to produce lactic acid, alcohols, acetic acid, glycerol, mannitol, and $CO_2$. Acetic acid bacteria oxidize alcohol to produce acetic acid; then, acetic acid is oxidized to produce $CO_2$ and water come out through the basket [2,6].

Temperature and pH changes occurred during cocoa bean fermentation. The initial temperature of fermentation was 29 °C. The increases in temperature were different in each treatment, ranging from 34 to 42 °C within 48–60 h. Then, there was a gradual decrease in temperature to 29 °C at the end of fermentation. The authors of [25] stated that the increase in the temperature of fermented cocoa beans from the ambient temperature (25–30 °C) to 35–40 °C within 48 h is due to the production of ethanol in an exothermic process. Research conducted by Ouattara et al. [29] also showed that the temperature was 29 °C at the beginning of the fermentation and reached a peak of 45 °C for 48–72 h with the addition of lactic acid bacteria. The pH value in the fermenting cocoa mass was in the range of 3.8–4.0 in the initial stage of fermentation (0–48 h). After 48 h, the pH increased to 4.5–7.5 until the end of fermentation.

The growth of the yeast, LAB, and AAB during the drying of the fermented cocoa beans decreased until the last day of drying. This growth agrees with the findings of research conducted by other authors [30]. This is because the drying process reduces the moisture content in the beans; thus, it is difficult for the microbes to grow.

The OTA content in the natural cocoa bean fermentation increased during the fermentation to a level of 4.97 µg/kg, although the sizes of the fungal populations decreased. This is because the OTA levels in cocoa beans accumulate and do not decrease during fermentation, although the sizes of the fungal populations decrease. The concentrations of mycotoxins in cocoa beans are retained in their shells [30]. There was an increase in the OTA levels during the drying of the naturally fermented cocoa beans to a level of 5.17 µg/kg at the end of drying. The increase in OTA levels was consistent with the increase in the sizes of the mold populations during drying (5.11 Log CFU/g on day 10).

The high OTA levels were related to the amount of *A. niger* YAC-9 added to the sample as a contaminant. The OTA level decreased slightly at the end of fermentation but increased at the end of the drying. This is consistent with the growth patterns of the fungi in this treatment. The fungi decreased in population until the end of fermentation, but when drying, the population of fungi continued to increase. The addition of the *A. niger* YAC-9 starter increased the size of the fungal population and the OTA levels in the cocoa beans. *A. niger* YAC-9 is known to be able to produce 57.68 ppb of OTA [23].

The use of the indigenous *L. plantarum* HL-15 as a starter culture individually or in combination with *C. famata* HY-37 and *Acetobacter* spp. HA-37 contributed to the reduction of the *A. niger* YAC-9 population and the OTA concentration in fermented cocoa beans. The results of our study prove that the use of yeast, LAB, and AAB cultures in the fermentation of cocoa beans can inhibit the growth of mycotoxin-producing fungi. Thus, the cocoa beans produced will be of a high quality and will have a high level of safety. The ability of LAB to bind with mycotoxins or reduce mycotoxin levels was also found in other studies [6,8,16]. Ngang et al. [28] found that the addition of an LAB isolate of A19 (*Pediococcus damnosus*) as a starter in cocoa fermentation with OTA-producing molds (*Aspergillus niger* and *Aspergillus carbonarius*) could inhibit the growth of the molds until they were almost undetectable, and the toxin production was reduced by 99%. Blagojev et al. [31] mentioned several possible mechanisms or interactions between lactic acid bacteria and mycotoxins, such as the inhibition of mycotoxin biosynthesis, the binding of mycotoxins to bacterial cell walls, or the detoxification of mycotoxins. The accumulation OTA in fermented cocoa beans can also be suppressed through the formation of organic acids during fermentation [32]. *Aspergillus niger* contamination and the synthesis of OTA during the fermentation of cocoa beans can be inhibited by using *C. famata* HY-37, *L. plantarum* HL-15, and *Acetobacter* spp. HA-37 as a starter culture such that the cocoa beans produced are of a good quality and are safe.

## 5. Conclusions

We found that the use of indigenous *L. plantarum* HL-15 either individually or in combination with *C. famata* HY-37 and *Acetobacter* spp. HA-37 as a culture starter in the fermentation process of cocoa beans can inhibit the growth of *A. niger* YAC-9 and OTA synthesis. Based on this, it is recommended that LAB, especially *L. plantarum* HL-15, be used in the fermentation of cocoa beans to improve the quality. This research must be continued in order to elucidate the roles of the three indigenous microbes in determining the quality of the flavor of fermented cocoa beans.

## 6. Patents

a.　Patent: Dry Starter Manufacturing Process of *L. plantarum* HLM-15 for Mycotoxin-Producing Fungus Control, 2017 (IDS000001851) (in Indonesian)

b.　Patent Cocoa Bean Fermentation Process with the Addition of Starter *L. plantarum* HL-15 to Inhibit Fungal Growth, 2019 (IDS000002554) (in Indonesian)

**Author Contributions:** E.S.R., T.F.D., T.M., and T.U. carried out the conceptualization and developed the methodology; R.T. and R.N.B.K. performed the experiments and analyzed the data. E.S.R., T.F.D., T.M., and T.U. wrote and reviewed the original manuscript; T.F.D., T.M., and R.U.H. wrote and edited the manuscript. All authors have read and agreed to the published version of the manuscript.

**Funding:** This research was funded by the Indonesian Agricultural Research and Development Agency (IAARD) with the contract numbers 54.19/H.M.240/I.1/03/2016.K (7th March 2016) and 55/60/HM.240/H.1/03/2017.K (20th March 2017).

**Institutional Review Board Statement:** Not applicable.

**Informed Consent Statement:** Not applicable.

**Data Availability Statement:** The data presented in this study are available on request from the corresponding author.

**Acknowledgments:** The authors thank the Indonesian Agricultural Agency Research Development (IAARD) for their support through the Project with contract number 54.19 /HM.240/I.1/3/2016.K, 2016, and 55.60/HM.240/H.1/03/2017.K, 2017. Thanks also goes to Sikstus Gusli for providing writing assistance.

**Conflicts of Interest:** The authors declare no conflict of interest.

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
