# Peer review of "Indigenous Yeast, Lactic Acid Bacteria, and Acetic Acid Bacteria from Cocoa Bean Fermentation in Indonesia Can Inhibit Fungal-Growth-Producing Mycotoxins"

_fermentation, doi:10.3390/fermentation7030192_

Round 1

Reviewer 1 Report

Dear Authors,

The manuscript ID: fermentation-1372586-v1 entitled “Indigenous Yeast, Lactic Acid and Acetic Acid Bacteria from Cocoa Bean Fermentation in Indonesia Can Inhibit Fungi Growth-Producing Mycotoxin” written by Endang Rahayu, Rokhmat Triyadi, Rosyida Khusna, Titiek Djaafar, Tyas Utami, Tri Marwati, and Retno Hatmi is devoted to cocoa beans fermentation.

The whole manuscript (Introduction, Materials and Methods, Results, Discussion and Conclusions) is properly organized. Introduction contains general data on fermentation of cocoa bean. The purpose of the work is concise and concrete. Appropriate methods and strains were used to perform these studies. Results are documented, presented in the form of figures, and right interpreted. Based on the results, adequate conclusions were drawn. With biological methods using indigenous Lactobacillus plantarum HL-15 individual or combination with Candida famata HY-37 and Acetobacter spp. HA-37, it was possible to effectively inhibit contamination with ochratoxin A-producing fungi. These microorganisms should be used in cocoa bean fermentation to inhibit the growth of the fungus-producing mycotoxins and improve quality. I agree with the Authors that this research needs to be continued to find out the role of the three indigenous microbes on the quality of fermented cocoa bean flavor. It is a well written and original article.

But, I have some suggestions in order to improve paper, which are the following:

Line 93: 109 CFU/mL – 109 CFU/mL

Lines 247-248: This is in line with research were conducted by [5,30]. – This is in line with research were conducted by others authors [5,30].

Line 260: Research conducted by [31] – Research conducted by Ouattara et al. [31]

Line 266: research conducted by [32]. – research conducted by other authors [32].

Line 290: used [6,8,16]. [33] mentions – used [6,8,16,33] mentions

Line 294: during fermentation[34].during fermentation [34].

Figure 7: In the Figure description, please correct the order (c), (b), (c), and (d) - into (a), (b), (c), and (d)

In the whole text: Acetobacter sppAcetobacter spp.

I think that this article is valuable and worth publishing in “Fermentation”, after minor review.

With highest regards,

Author Response

Dear Reviewer 1, our responses stated in the the attachment. Please kindly check it.

Thank you.

Reviewer 2 Report

Fermentation review20210903

This manuscript describes the fermentation profiles of yeast, lactic acid bacteria, acetic acid bacteria and aspergillus niger in cocoa bean. They have shown that by incubating with lactic acid bacteria, ochratoxin production can be decreased.

This ms is well described and research is clear cut.

However, several points should be addressed

English grammar is far from satisfactory . English editing is needed

Preceding researches on ochratoxin production in cocoa and lactic acid bacteria should be cited and discussed.

Author Response

Dear Reviewer 2, our responses are stated in the attachment. Please check it.

Thank you. 
